# Deep learning workflow to support in-flight processing of digital aerial imagery for wildlife population surveys

Tsung-Wei Ke[1]*, Stella X. Yu[1,2]*, Mark D. Koneff[3]*, David L. Fronczak[4], Luke J. Fara[5], Travis J. Harrison[5], Kyle L. Landolt[5], Enrika J. Hlavacek[5], Brian R. Lubinski[4], Timothy P. White[6]

**1** University of California Berkeley, Berkeley, California, United States of America, **2** University of Michigan, Ann Arbor, Michigan, United States of America, **3** Division of Migratory Bird Management, United States Fish and Wildlife Service, Orono, Maine, United States of America, **4** Division of Migratory Bird Management, United States Fish and Wildlife Service, Bloomington, Minnesota, United States of America, **5** Upper Midwest Environmental Sciences Center, United States Geological Survey, La Crosse, Wisconsin, Minnesota, United States of America, **6** Environmental Studies Program, Bureau of Ocean Energy Management, Sterling, Virginia, United States of America

* twke@berkeley.edu (TWK); stellayu@berkeley.edu (SXY); mark_koneff@fws.gov (MDK)

**Data Availability Statement:** The data associated with this publication is provided through the U.S. Geological Survey's ScienceBase Catalog (https://

## Abstract

Deep learning shows promise for automating detection and classification of wildlife from digital aerial imagery to support cost-efficient remote sensing solutions for wildlife population monitoring. To support in-flight orthorectification and machine learning processing to detect and classify wildlife from imagery in near real-time, we evaluated deep learning methods that address hardware limitations and the need for processing efficiencies to support the envisioned in-flight workflow. We developed an annotated dataset for a suite of marine birds from high-resolution digital aerial imagery collected over open water environments to train the models. The proposed 3-stage workflow for automated, in-flight data processing includes: 1) image filtering based on the probability of any bird occurrence, 2) bird instance detection, and 3) bird instance classification. For image filtering, we compared the performance of a binary classifier with Mask Region-based Convolutional Neural Network (Mask R-CNN) as a means of sub-setting large volumes of imagery based on the probability of at least one bird occurrence in an image. On both the validation and test datasets, the binary classifier achieved higher performance than Mask R-CNN for predicting bird occurrence at the image-level. We recommend the binary classifier over Mask R-CNN for workflow first-stage filtering. For bird instance detection, we leveraged Mask R-CNN as our detection framework and proposed an iterative refinement method to bootstrap our predicted detections from loose ground-truth annotations. We also discuss future work to address the taxonomic classification phase of the envisioned workflow.

## Introduction

Natural resource management agencies in North America have a long history of using crewed aircraft to monitor marine and terrestrial wildlife populations. The data collected

www.sciencebase.gov/catalog/) and can be accessed at doi:10.5066/P9CBZQV1.

**Funding:** SXY, Fireball International Services Corporation grants 17UCR-1751 and 19UCR-2020 (included funds from US Fish and Wildlife Service contract 140F0918P0139), http://fireball.international, http://www.fws.gov, USFWS staff collaborated on all phases of research. SXY, U.S. Geological Survey Cooperative Agreement #G19AC00203, http://www.usgs.gov, USGS staff collaborated to all phases of this research.

**Competing interests:** The authors have declared that no competing interests exist.

during these surveys have an important role in informing agency regulatory, management, and conservation decision processes [1–5]. Advanced remote sensing technologies combined with automation through machine learning and computer vision may improve the safety and quality of aerial survey data collections [6]. Aircraft are often used to survey conspicuous wildlife species in remote areas with limited ground accessibility or when large areas must be sampled cost-efficiently. The U.S. Fish and Wildlife Service and Canadian Wildlife Service survey waterfowl populations over a large portion of North America's upland and marine environments to inform migratory bird harvest regulation decisions and to focus habitat conservation. Similarly, the Bureau of Ocean Energy Management, National Oceanic and Atmospheric Administration, and other agencies invest heavily in population monitoring of marine wildlife to guide offshore conservation and management rooted in minimizing potential disturbance and displacement effects associated with energy development on the Outer Continental Shelf (OCS).

Wildlife population surveys are often conducted by human observers making visual counts from low flying aircraft (45–61 meters above ground level, AGL) or manually processing remote sensing data acquired at higher altitudes. While low-level, visual surveys can be successful and cost-efficient in supporting agency decision-making, they subject agency personnel to substantial risk. Aviation accidents are the leading cause of on-the-job fatalities among wildlife biologists in the U.S. [7]. Also, visual surveys, involving multiple air-crews and observers, must include methods to minimize or estimate important biases, including those known to vary widely among observers such as detection, misclassification, group-size estimation, and sample area determination [8–14]. Wildlife population estimates and crew safety will benefit from greater integration of airborne remote sensing as a primary data collection tool and as a means of estimating and correcting for visual survey biases.

A major challenge to the integration of remote sensing methods for large-scale population surveys is the tremendous volume of data that is collected during image-based surveys and the lack of efficient tools for automated and rapid detection, classification, and counting of wildlife targets. In some cases, agency wildlife population surveys collect data on dozens of species concurrently, are regional or continental in scope, involve simultaneous participation of multiple air-crews, and can extend for several weeks [15–17]. High spatial resolution imagery is also required for identification of smaller species such as birds. Such broad-scale, high-resolution surveys, if implemented using remote sensing methods, would generate hundreds, or even thousands, of terabytes of image data per survey event. Automation of wildlife detection and classification from imagery is critical if remote sensing solutions are to be cost-efficient for natural resource management agencies at broad scales. While many analytical approaches have been investigated to support cost-efficient remote sensing solutions for wildlife population monitoring [18–22], machine learning, specifically deep learning [23–28], shows great promise for automating detection and classification of wildlife from digital imagery [29–33].

The objectives of this study focused on two stages of an envisioned in-flight workflow: 1) develop and evaluate automated methods to rapidly filter and subset digital aerial imagery based on the probability of marine bird presence at the image-level and 2) detect and mark individual occurrences of marine birds on images with a high probability of occurrence. We also evaluated the performance of both the filter and detection algorithms in relation to image resolution. We focused initially on several sea duck species that are hunted and species of management concern [34–37] including black (*Melanitta americana*), surf (*M. perspicillata*), and white-winged scoter (*M. deglandi*), common eider (*Somateria mollissima*), and long-tailed duck (*Clangula hyemalis*). While there has been considerable research attention directed at detection [1, 29, 31] and classification [2, 38] of birds and other wildlife from

imagery through deep learning techniques, incorporating processing efficiencies such as image sub-setting or filtering into the workflow supports development of in-flight processing and efficient collection of wildlife population data over broad geographic scales through remote sensing.

## Methods

### Study area

Our study used aerial imagery of the Nantucket Shoals, Massachusetts, USA, and Lake Michigan near Manitowoc, Wisconsin, USA (Fig 1) acquired by the U.S. Fish and Wildlife Service [39]. Imagery was acquired in February 2017 and October 2016 for the Nantucket Shoals and for Lake Michigan, respectively. The Nantucket Shoals area is a shallow bank with bathymetric, substrate, and tidal characteristics that concentrate prey favored by wintering marine birds and the area can harbor large aggregations annually [5, 17, 40]. Large concentrations of long-tailed ducks winter on Lake Michigan [41–44] including the region near Manitowoc County, Wisconsin [45].

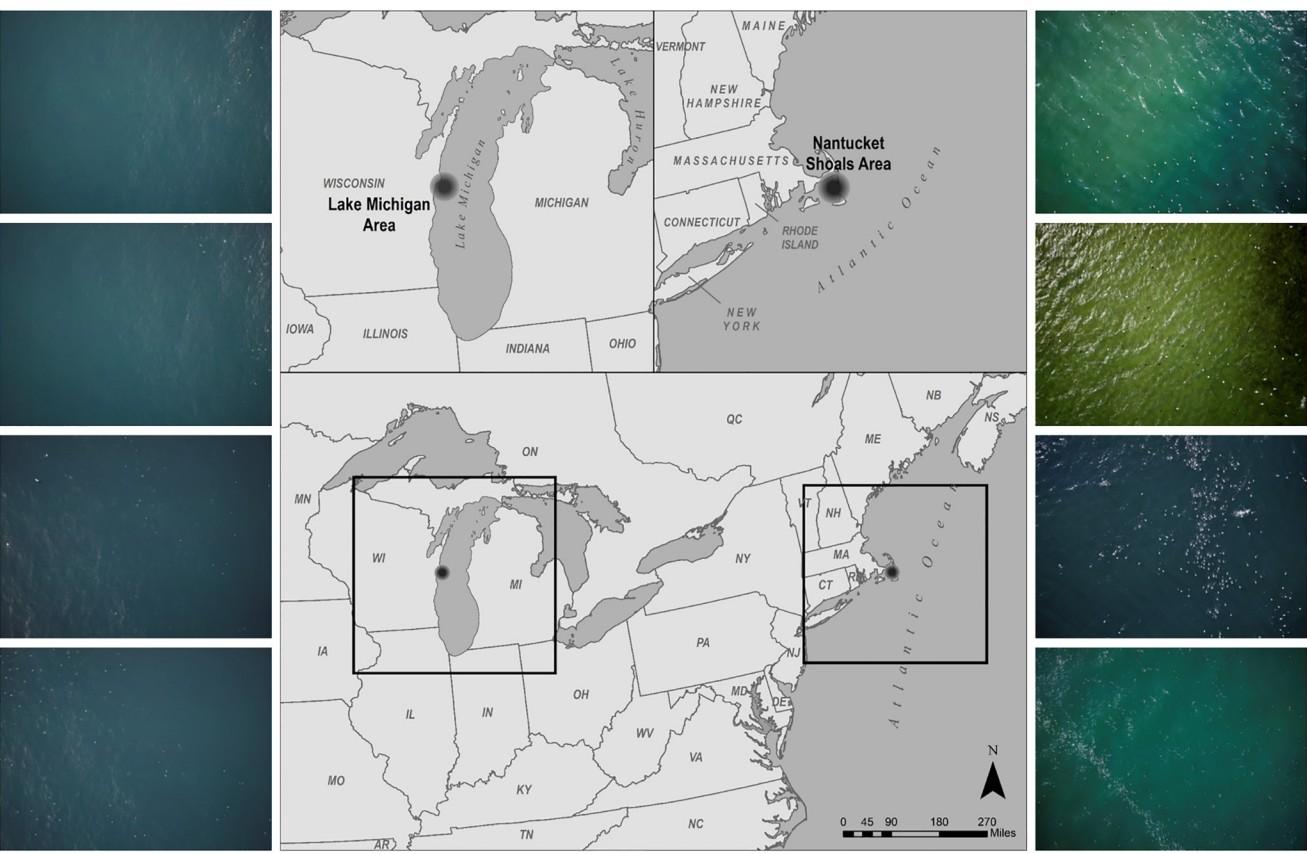

**Fig 1. Imagery used in this study was collected over Lake Michigan, near Manitowoc, Wisconsin, and the Nantucket Shoals, Massachusetts.** The images on the left show samples of imagery collected from Lake Michigan and the images on the right were collected from the Nantucket Shoals. The authors created the map figure using publicly accessible data with no use constraints. The source information for the map data used in the figure is covered by the Geogratis License Agreement for Unrestricted Use of Digital Data. No base map or other copyrighted material was used to create the map figure. Source information: Commission for Environmental Cooperation (CEC). North American Atlas—Political Boundaries. [shapefile]. 2010. Montreal, Quebec: Government of Canada. Available via Commission for Environmental Cooperation: http://www.cec.org/north-american-environmental-atlas/political-boundaries-2010/.

## Imagery acquisition

All imagery was acquired from a Partenavia P68 fixed-wing airplane using a PhaseOne iXU-R 180 forward motion compensating 80-megapixel digital frame camera with a 70 mm Rodenstock lens. The PhaseOne sensor was integrated with a Global Positioning System and Internal Navigational System in a direct georeferencing system capable of estimating frame-specific exterior orientation parameters necessary for image orthorectification without ground control. The charge-coupled device imager for the PhaseOne camera was $10328 \times 7768$ pixels. Altitudes of acquisition were between 24.4 to 198.1 m for the Nantucket Shoals site and 21.3 to 42.7 m for the Lake Michigan site. Ground sample distance (i.e., pixel resolution) ranged from 0.18 to 1.47 cm for the Nantucket Shoals site and from 0.14 to 0.32 cm for the Lake Michigan site. The wind speeds during the flights were below 20 knot. The sky conditions varied from clear to broken leading to areas of glare. Sea surface varied from near flat to moderate wave action. Imagery was collected in Phase One's proprietary IIQ raw image format and converted to color-balanced TIFF images using Phase One's Capture One image processing software.

## Training data annotation

Deep learning models iterate on training datasets to estimate their parameters for inference on new data. To prepare labeled training data needed for the development of deep learning models, experienced waterfowl aerial observers (from the U.S. Fish and Wildlife Service and the U. S. Geological Survey) annotated birds in the study's imagery dataset. Using ArcGIS, annotators manually drew bounding boxes around birds and labeled each bird instance with relevant species and cohort based attributes. All annotated bird instances had to be clearly visible to the annotator; birds submerged or partially covered by water were not annotated. While the primary objective of this study was detection of bird instances in an image, we believed that a training dataset that captured variability in appearance of both birds and imagery background would improve detection performance and also aid in subsequent classification phases.

Because our primary interest was in the development of an efficient, in-flight workflow and initial efforts focused on image filtering and individual target detection, we did not cross-check annotations among multiple observers, rather we treated each annotators work as error-free. Operationally, we recognize the importance of error at the annotation/training set development stage and will assign each annotated image to at least two analysts. The annotations of the two analysts will be compared and discrepancies will be resolved in consultation with a third independent taxonomic expert.

Manually-drawn bounding boxes were then transformed into 4-vertex, axis-aligned rectangles. In an effort to improve detection results, we used an iterative bootstrapping method (Fig 2) to adjust the original bounding boxes to more tightly surround each bird instance. Specifically, we used a convolutional activation map (CAM), often employed for object localization [46, 47], and the level-set algorithm to tighten the manually-created annotation boxes around the bird instances.

## Data preprocessing

We sub-divided the imagery dataset into three sets: a training dataset, a validation dataset, and a test dataset. The training dataset was used to provide examples for training the model and fitting the hyper-parameters (e.g., learning rate, momentum, epoch, etc.). These hyper-parameters modified the model training process to improve model accuracy and efficiency. The validation dataset reported how well the model performed during training and was used to optimize the hyper-parameter settings. The test dataset was separated completely from the training and validation process and used for final model evaluation. Our training and

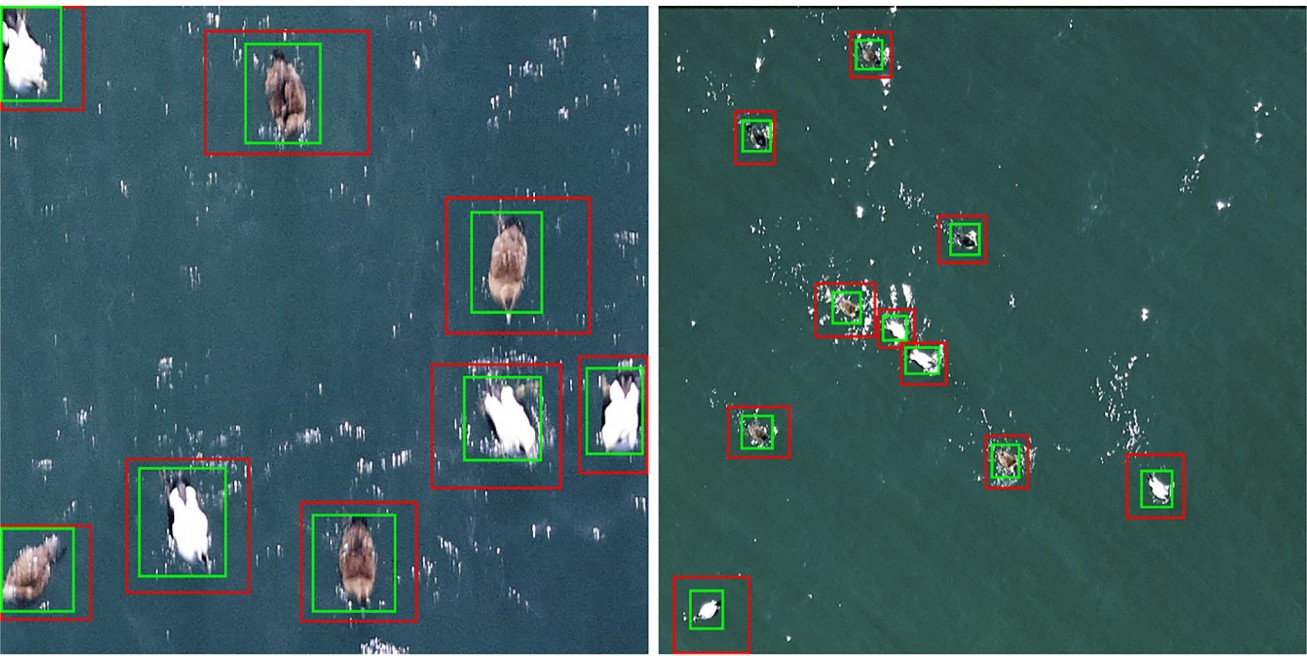

**Fig 2. A bootstrapping method was used to refine manual annotation boxes (red) to more tightly encompass bird instances (green boxes).**

validation datasets were prepared by randomly splitting images from the Nantucket Shoals study area into two sets of 38 and 20 images, respectively. Nine images from the Lake Michigan study area were used as the test dataset (S1 Table). Using the Lake Michigan study area for the test dataset allowed us to evaluate how robust the model generalizes across images from different survey areas.

Computer hardware limitations precluded processing of full parent images acquired by the PhaseOne camera used in this study (images consist of 80 megapixels; 10328 × 7768 pixels). We cropped images into smaller patches to reduce computational memory usage. Training images were cropped into patches of 720 × 720 pixels at every sliding window location, and the step size of the sliding window was set to 540 × 540. For the validation and test datasets, the images were uniformly cropped into patches of 1440 × 1440 pixels with a stride of 1080 × 1080. The process of cropping images generated empty, water-only patches, which were needed in the training and validation datasets to ensure the model could also identify instances where no birds were present. It is important to note that each bird instance may have repeatedly appeared in different patches; these duplicates were considered independent instances for simplification. The implications of potential double-counting of bird instances for population survey objectives are discussed later. We used non-maximum suppression to remove boxes that overlapped significantly, which reduced the number of double-counted birds.

## Automated workflow for imagery processing

We proposed an automated, in-flight workflow to address challenges to broad-scale wildlife population monitoring using remote sensing technologies. As shown in Fig 3, our envisioned workflow consisted of: 1) rapidly filtering imagery into subsets based on the probability of a bird occurrence, 2) detecting bird instances, and 3) classifying bird species. For the first

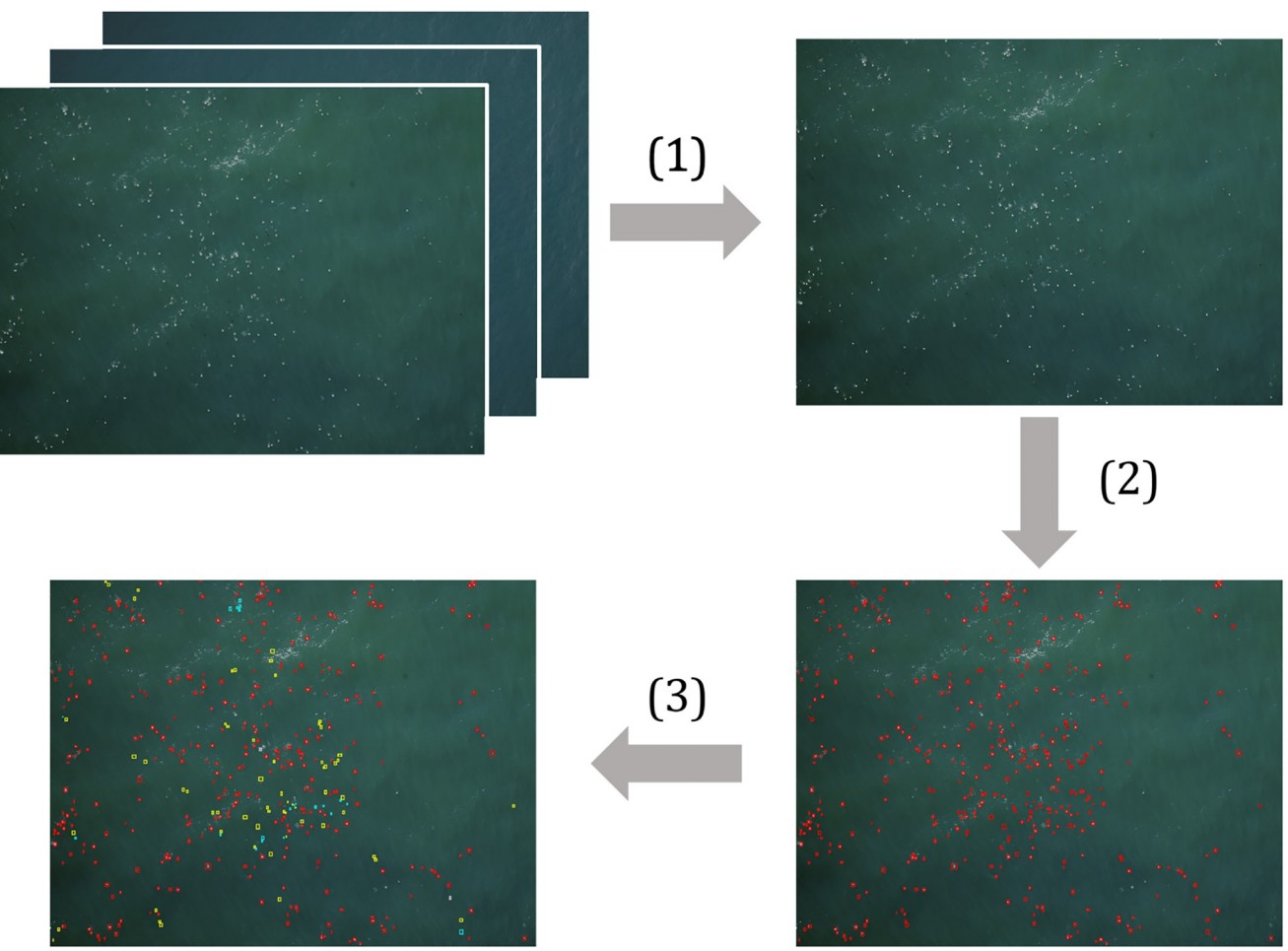

**Fig 3. A proposed 3-stage workflow for efficient imagery filtering, detection, and classification of bird species.** (1) Images were first subset based on the probability of a bird occurrence to filter out empty, water-only images. (2) The set of images for which probability of occurrence exceeded a threshold value were then processed by a detection model to generate a bounding box around each instance. (3) Bird instances would then be labelled with fine-grained species identifiers by a classification model (represented by different colored boxes) in this as yet undeveloped stage of the workflow.

stage, we used a classification model to categorize each image as containing a bird instance or not containing a bird instance. For images with a probability of occurrence greater than a pre-defined threshold, we applied a detection model to mark and annotate birds. In the final, as yet undeveloped, stage in the proposed workflow, we would categorize detected bird instances into fine-grained classes (e.g., species). In this paper, we focused on the first two steps (image filtering and bird detection) and left the last step (species classification) as future work.

**Image filtering.** The first stage of our envisioned workflow was the application of an algorithm to filter or subset imagery based on the probability of bird occurrence at the image level. Rapidly screening out images that have a low probability of bird occurrence increases processing efficiency by reducing the volume of data passed through subsequent in-flight workflow stages. The selected probability threshold should balance objectives for accuracy and rare-species detection with in-flight processing efficiency. All imagery would be retained, including images not subjected to further in-flight processing, in order to estimate error rates of the filter and to support future, as yet unidentified, applications.

As this is generally considered a binary classification problem, we used a model built upon ResNet18 [48], which is composed of 17 convolutional layers and one fully connected layer. We also evaluated and contrasted the performance of Mask R-CNN [49] as a first-stage filter. The Mask R-CNN algorithm is described in greater detail in the following section on Bird Detection. We followed a similar data augmentation process as in [48] to enlarge our training set. We randomly cropped patches from each 720 × 720 pixel image, which are 0.08–1.0 times image size. We then re-scaled the cropped patches to 480 × 480 pixels and randomly flipped the image horizontally. For each cropped patch, we labeled the patch if it included a bird instance. For test images, we re-scaled the image from 1440 × 1440 into 960 × 960 pixels, reducing the compute time without jeopardizing the classification performance. When the model was run on the validation and test set of images, we aggregated the patches into their original parent image size to output the binary predictions of empty versus non-empty parent images. If any patch from a parent image was predicted to contain an object, then the entire parent image was labeled as containing an object.

**Bird detection.** The second stage of the processing workflow focused on generating bounding boxes around each detected bird instance. The objective was to predict the "objectness" of a region and produce corresponding bounding box coordinates. We used a Support Vector Machine (SVM) model as a baseline for comparison with the Mask R-CNN [49] model, since SVM was considered state-of-the art prior to deep learning approaches.

Once the data was pre-processed, bounding box proposals were generated using the processing workflow depicted in Fig 4. The image was binarized by converting the pixels to black or white and applying a dilated convolution. We set the intensity threshold value to 0.9 and dilation factor to 12. We also enhanced the contrast before thresholding the image. We were left with binary blobs that we could segment as different instances by labeling connected components. We labeled the connected components by 8-pixel connectivity. We derived bounding boxes surrounding each blob instance by using the minimum and maximum x and y coordinates of the blob. We then extracted the Histogram of Oriented Gradients (HOG) features from each bounding box proposal and trained an SVM [50] model to predict if the box contained a bird instance.

The Mask R-CNN model is a deep neural network designed for detecting objects in an image [49]. The Mask R-CNN architecture functions in a two step process: the first step generates broad proposals of where an object might be found in the image and the second step places a refined bounding box and mask on the proposed object. We adopted Resnet50 [48] and FPN [51] as the convolutional backbone; the Resnet50 model was initiated from Imagenet-pretrained weights [52].

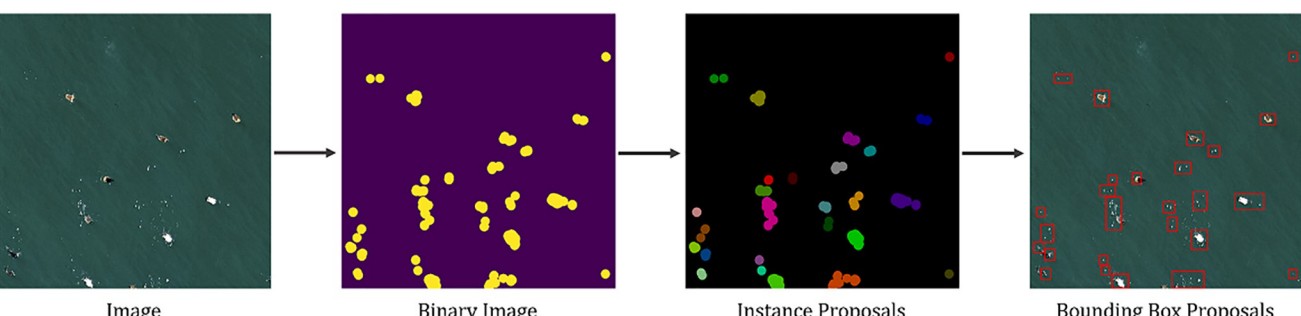

| Image | Binary Image | Instance Proposals | Bounding Box Proposals |

**Fig 4. Processing workflow for bird detection.** Bounding box proposals of SVM Baseline were generated by thresholding and dilating the input image into binary blobs, labeling the connecting components as instance proposals, labeling instance proposals by 8-pixel connectivity, and deriving bounding box proposals from the minimum and maximum x and y coordinates of each labeled blob segment.

We implemented the Mask R-CNN model provided by [53] by ingesting the refined bounding boxes (from our bootstrapping process described earlier) as a bird instance. For training the Mask R-CNN, we set the hyper-parameters as follows: input resolution to $720 \times 720$, epochs to 48, batch size to 16, and momentum to 0.9. We set the initial learning rate to 0.02 and decreased the learning rate by 0.1 at epoch 32 and 44. We randomly re-scaled and horizontally flipped the images for data augmentation. We did not augment the data for testing purposes.

For evaluating the universality of Mask R-CNN detection algorithm, we conducted cross-validation experiments by using imagery acquired from the different study areas for training. We trained two separate Mask R-CNN models using Nantucket Shoals and Lake Michigan areas (train and test dataset), and evaluated the detection performance over Nantucket Shoals images of validation dataset.

### Effects of image resolution

Imagery used in this study was acquired under various environmental conditions from different locations and altitudes. Varying collection altitudes resulted in different image resolutions, which affected the size of bird targets in image space and the level of fine detail observable for individual bird instances (Fig 5). We evaluated the effect of image resolution on the performance of the preferred image filter (binary classifier) and detection (Mask R-CNN) models. Our images were captured at different altitudes and corresponding ground sample distances (GSD), ranging from 0.14 to 1.47 cm (Fig 5). To evaluate the effect of image resolution on performance, this analysis was conducted on the validation image set. For the validation image set, we computed the histogram of image GSD and, by visual examination, split our validation dataset into three groups corresponding to ground sample distances of < 0.6 cm, 0.6–1.2 cm, and > 1.2 cm.

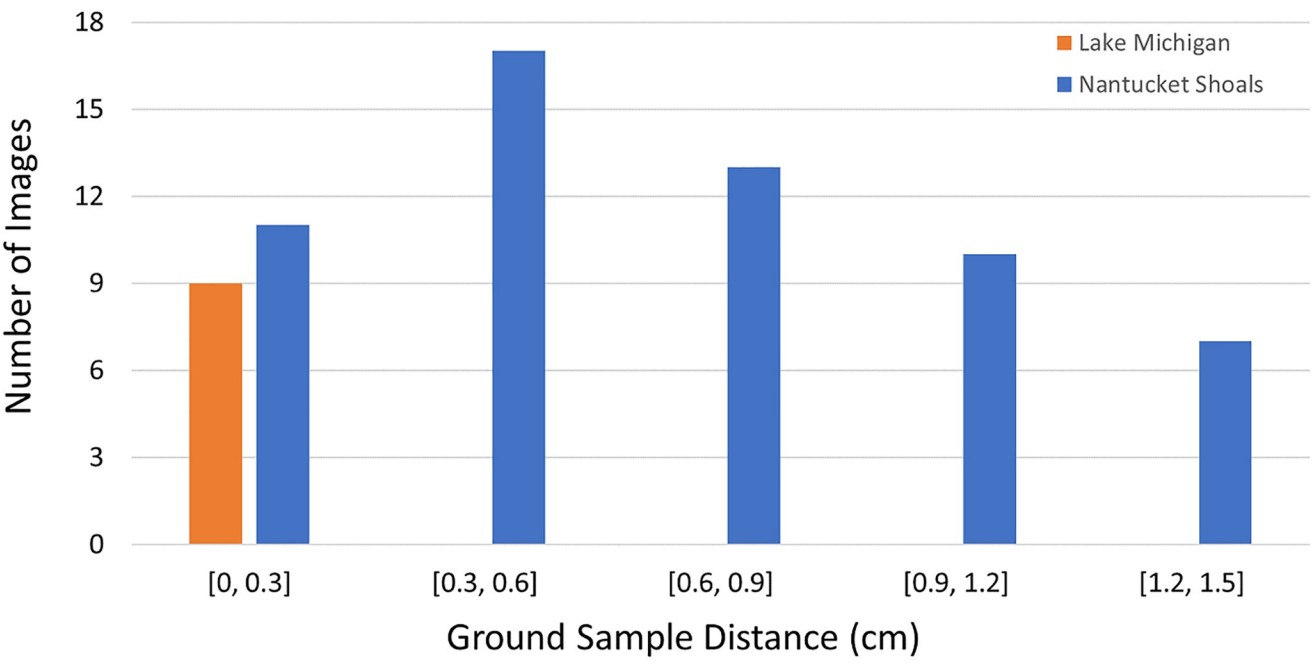

**Fig 5. Distribution of image resolutions in the Nantucket Shoals and Lake Michigan imagery datasets.**

## Performance metrics

We reported the performance on both validation and test sets. Performance of the validation set indicated how well the model fit the training data, whereas the test set performance indicated how generalizable the model was to the test set.

We used recall (R), precision (P), and accuracy to evaluate the performance of the binary classification filter. To evaluate the performance of the detection model, we computed Average Precision (AP) and Average Recall (AR) [54]. AP evaluates the precision of ranked detection results by measuring the area under the precision-recall curve computed from ranked outputs. AP is formulated as

$$AP = \sum_r p(r)$$ (1)

where $p(r)$ is the precision at recall $r$, and $r$ ranges from 0 to 1 with step size 0.01. AR is the average of recall over different query conditions.

The intersection over union (IoU) metric computed for bounding box predictions in comparison to ground-truth annotation boxes is used to specify a true positive detection. Specifically, $IoU = \frac{area(B_p \cap B_{gt})}{area(B_p \cup B_{gt})}$, in which $B_p$ and $B_{gt}$ denote predicted and ground-truth bounding boxes. Following [54], we calculated P, R, and AP under different IoU threshold values, ranging from 0.5 to 0.95 with a step size of 0.05. IoU threshold values, specified by the analyst, control how strict the correspondence between predicted and ground-truth boxes must be for the prediction to be classified as a true positive. The IoU metric is stricter at higher threshold values, which require the prediction to more closely align with ground-truth annotations before a true positive detection is confirmed (Fig 6). In contrasting model performance we reported mean AP (mAP) and AR at threshold values of 0.50 and 0.75.

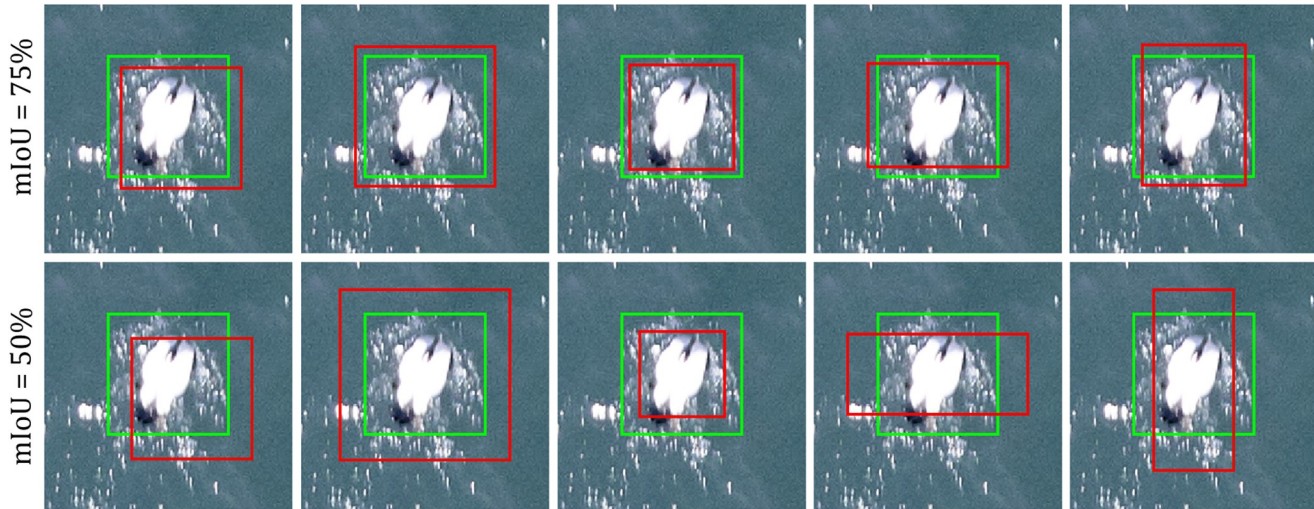

**Fig 6. True positive detections with different IoUs.** Higher intersection over union (IoU) values correspond to a tighter correspondence between predicted bounding boxes (red) and manually-derived ground-truth boxes (green). Higher IoU thresholds demand a greater correspondence between prediction and ground truth to classify a prediction as a true positive.

**Table 1. Performance of the binary classifier and Mask R-CNN models in predicting the occurrence of any bird instance(s) at the image level.**

| Method | Backbone | R | P | Acc. | R | P | Acc. |
|---|---|---|---|---|---|---|---|
| Binary Classifier | Resnet18 | 1.0 | 1.0 | 1.0 | 1.0 | 1.0 | 1.0 |
| Mask R-CNN | FPN-Resnet50 | 1.0 | 1.0 | 1.0 | 1.0 | 1.0 | 1.0 |

White and gray-colored backgrounds denote validation and test set results, respectively. **R** = recall; **P** = precision; and **Acc.** = accuracy.

## Results and discussion

### Image filtering

We evaluated our image filtering algorithms by aggregating the patches created during the machine learning workflow back into their original parent image size. If any patch in a parent image was predicted to contain an object, then the entire image was labeled as being occupied by a bird(s). For the purpose of image filtering, we evaluated the performance of our two candidate algorithms—the Resnet-18-based binary classifier and Mask R-CNN—in predicting the occurrence of any bird instance in an image (Table 1). On the validation and test dataset, both the binary classifier and Mask R-CNN model achieved high performance among three metrics (recall = 100%; precision = 100%; accuracy = 100%). Owing to its simpler architecture, the binary classifier performed faster than Mask R-CNN (with the computational ability to process 11.1 frames per second (fps), compared to 7.69 fps, respectively), an important consideration in our proposed in-flight workflow.

Because our validation and test image sets were all known to contain bird targets, we were concerned that our accuracy metrics were overly confident as these datasets did not contain a realistic distribution of empty versus non-empty bird occurrence at the scale of a parent image. To explore this, we evaluated the performance of the binary classifier and Mask R-CNN models on an independent set of 69 images, together with 9 images of test dataset (S2 Table) from the Lake Michigan study area that were manually annotated as either containing or not containing bird instances (Table 2). Using this set of imagery, the accuracy of the binary classifier was reduced to 95%, while Mask R-CNN achieved an accuracy of 35%.

### Bird detection

For the second stage of our envisioned workflow, we evaluated the performance of our candidate models—baseline SVM and Mask R-CNN—for detecting individual bird instances at the level of patches cropped from parent images during computation (Table 3). The SVM baseline performed poorly on our dataset, achieving mAP and AR of 1% and 2% respectively on the validation dataset. The Mask R-CNN method outperformed the baseline on all metrics. It achieved 29% and 42% for mAP and AR on the validation dataset using the initial, manually-

**Table 2. Performance of the binary classifier and Mask R-CNN models in predicting the occurrence of any bird instance(s) at the image level applied to 69 additional images together with 9 images of test dataset from Lake Michigan.**

| Method | Backbone | R | P | Acc. |
|---|---|---|---|---|
| Binary Classifier | Resnet18 | 0.95 | 0.86 | 0.95 |
| Mask R-CNN | FPN-Resnet50 | 0.95 | 0.26 | 0.35 |

These images were not fully annotated but were manually labeled as containing or not containing birds. **R** = recall; **P** = precision; and **Acc.** = accuracy.

**Table 3. Performance of the baseline SVM and Mask R-CNN detection algorithms based on the original, manually-derived annotated bounding boxes.**

|  | *mAP* | *mAP*$_{50}$ | *mAP*$_{75}$ | *AR* |
|---|---|---|---|---|
| SVM Baseline | 0.01 | 0.04 | 0.00 | 0.02 |
| Mask R-CNN | 0.29 | 0.75 | 0.13 | 0.42 |
| SVM Baseline | 0.02 | 0.11 | 0.00 | 0.05 |
| Mask R-CNN | 0.18 | 0.61 | 0.08 | 0.33 |

White and gray-colored backgrounds denote validation and test set results, respectively. **mAP** = mean average precision; **mAP$_{50}$** = mean average precision at IoU 50%; **mAP$_{75}$** = mean average precision at IoU 75%; **AR** = average recall.

**Table 4. Performance of the Mask R-CNN detection algorithm based on the original, manually-derived, annotation bounding boxes, and the refined annotation bounding boxes.**

|  | *mAP* | *mAP*$_{50}$ | *mAP*$_{75}$ | *AR* |
|---|---|---|---|---|
| Original Annotation | 0.29 | 0.75 | 0.13 | 0.42 |
| Refined Annotation | 0.56 | 0.93 | 0.65 | 0.65 |
| Original Annotation | 0.18 | 0.61 | 0.08 | 0.33 |
| Refined Annotation | 0.47 | 0.90 | 0.41 | 0.57 |

White and gray-colored backgrounds denote validation and test set results, respectively. **mAP** = mean average precision; **mAP$_{50}$** = mean average precision at IoU 50%; **mAP$_{75}$** = mean average precision at IoU 75%; **AR** = average recall.

derived annotated bounding boxes. Based on the original, manual annotations, a large performance discrepancy was apparent (62%) between mAP at the IoU threshold 0.5 and 0.75. This is an artifact of the original training data bounding boxes not being tightly constrained around bird targets and incorporating too many background pixels.

After applying the bootstrapping method to refine our annotated bounding boxes to more tightly encompass individual bird instances, we retrained the Mask R-CNN detection model on these data and contrasted performance with the model trained on the original, manual annotations. The refined annotated bounding boxes substantially improved the performance of the Mask R-CNN algorithm across all performance metrics (Table 4). On the validation set, the mAP increased from 29% to 56% and the AR increased from 42% to 65%. The performance gap between mAP at IoU = 0.5 (93%) and 0.75 (65%) was much smaller using the refined annotation data. Improved performance related to the refined annotation bounding boxes was consistent across all IoU thresholds (Fig 7). This performance improvement was apparent across both validation and test data sets.

We cross-validated the Mask R-CNN algorithm by using imagery acquired from the different study areas for training. We compared Mask R-CNN models trained with Nantucket Shoals (training set) and Lake Michigan (test set) imagery (Table 5). On the validation set, the mAP decreased from 47% to 15% and the AR decreased from 57% to 31%.

## Effects of image resolution

We also evaluated the performance of the binary classifier model (for the image filtering stage of the workflow) and the Mask R-CNN detection model (for the bird detection stage) across the range of image resolutions (i.e., pixel GSD) represented in our validation dataset by

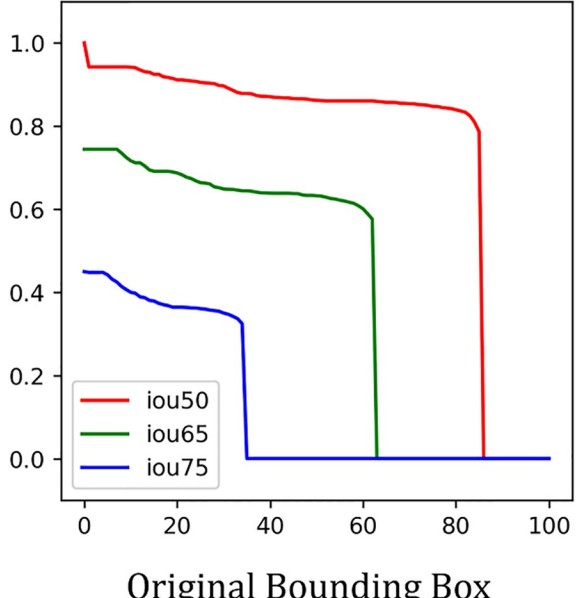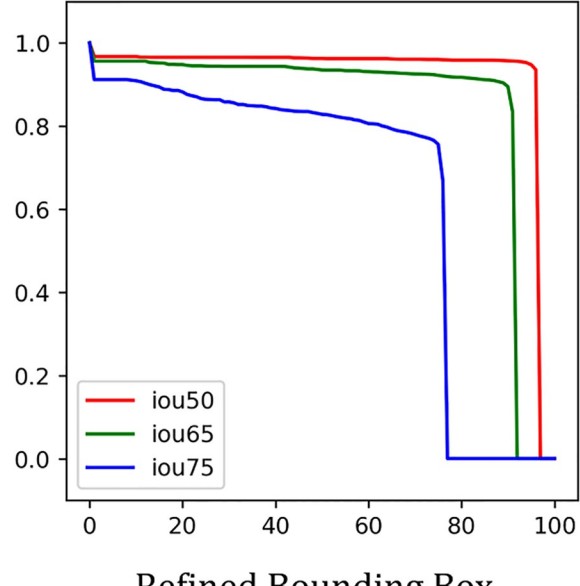

Original Bounding Box                Refined Bounding Box

**Fig 7. Precision-Recall curve for original, manually-derived, and refined annotation bounding boxes for the Mask R-CNN detection model applied to the validation imagery set.** As the IoU threshold increased, the detection model achieved better performance with refined annotations.

partitioning imagery into 3 subsets based on GSD: < 0.6 cm, 0.6–1.2 cm, and > 1.2 cm. The variation in GSD in the test data set was insufficient to evaluate the effect of image resolution on the test data. Using the validation data set, we found no substantial effect of image resolution on the performance of the binary classifier model (for image filtering) in predicting the occurrence of a bird target(s) at the image level (Table 6). The binary classifier model achieved accuracy between 97–99% over the three levels of image resolution.

We also evaluated the performance of the Mask R-CNN detection model across the range of GSD values in our image set (Table 7). Detection performance was strongly affected by image resolution. As ground sample distance increased from < 0.6 cm to > 1.2 cm, mAP decreased from 63% to 46%. The performance gap was more significant when mAP was defined at a higher IoU. At IoU = 0.75, mAP declined from 77% to 44% as pixel GSD increased.

## Conclusions and future directions

In this study, we evaluated the performance of deep learning models within a proposed in-flight workflow to support efficient monitoring of wildlife populations over broad geographic

**Table 5. Effect of training regions on the performance of the Mask R-CNN detection algorithm based on the refined annotation bounding boxes.**

|  | *mAP* | *mAP$_{50}$* | *mAP$_{75}$* | *AR* |
|---|---|---|---|---|
| Nantucket Shoals | 0.47 | 0.90 | 0.41 | 0.57 |
| Lake Michigan | 0.15 | 0.49 | 0.06 | 0.31 |

The Mask R-CNN model is cross-validated using images at Nantucket Shoals and Lake Michigan for training. The results are reported based on validation set at Nantucket Shoals. **mAP** = mean average precision; **mAP$_{50}$** = mean average precision at IoU 50%; **mAP$_{75}$** = mean average precision at IoU 75%; **AR** = average recall.

**Table 6. Effect of image resolution on the performance of the binary classifier algorithm in predicting occurrence of a bird target(s) on an individual image using the validation imagery set.**

| GSD (cm) | Recall | Precision | Accuracy |
|---|---|---|---|
| < 0.6 | 0.98 | 0.97 | 0.97 |
| 0.6–1.2 | 0.98 | 0.98 | 0.99 |
| > 1.2 | 0.98 | 0.98 | 0.98 |

**Table 7. Effect of image resolution on the performance of the Mask R-CNN algorithm in detecting individual bird instances as applied to the validation imagery set.**

| GSD (cm) | mAP | $mAP_{50}$ | $mAP_{75}$ | AR |
|---|---|---|---|---|
| < 0.6 | 0.63 | 0.95 | 0.77 | 0.71 |
| 0.6–1.2 | 0.59 | 0.97 | 0.69 | 0.67 |
| > 1.2 | 0.46 | 0.88 | 0.44 | 0.55 |

**mAP** = mean average precision; $mAP_{50}$ = mean average precision at IoU 50%; $mAP_{75}$ = mean average precision at IoU 75%; and **AR** = average recall.

regions using remote sensing technology. Specifically, we evaluated 1) a binary classifier algorithm as a means to filter images based on predicted bird occurrence and 2) a Mask R-CNN algorithm for detecting individual marine bird instances. We also assessed whether image resolution affected the performance of either filtering or detection algorithms.

Once the training set annotation bounding boxes were refined to more closely align with individual bird targets, the preferred binary classifier and Mask R-CNN detection algorithms performed well, generating high precision and accuracy. Additional effort to optimize hyperparameters should further improve performance. As a first-stage image filter, the binary classifier model was computationally more efficient due to its simpler architecture, an important consideration for our in-flight processing workflow.

The algorithms showed some promise for universal open-water environments based on the results from the independent test dataset from Lake Michigan. However, the cross-validation results and variable performance of the detection algorithm when training and testing image sets were transposed indicate the need for a more spatially and temporally robust training data set to improve universality. Image filter algorithm performance was largely unaffected by the ground sample distance over the range of variability expressed in our data. However, we observed significant effects in the performance of the detection model across the range of image resolutions included in our dataset. The performance degraded as image GSD increased. Additional targeted imagery collection will be necessary to evaluate trade-offs in detection performance versus image resolution and acquisition costs. These trade-offs will also have to be re-evaluated as development focus shifts from detection to taxonomic classification.

We also recognize that our imagery set is limited in geographic coverage and represents a relatively narrow range of environmental variability. Filter and detection algorithm performance can be expected to be influenced by more extreme environmental conditions such as increased sea-state or low-light due to time-of-day or cloud cover. Low-light causes loss of contrast and target detail, increased sea-state adds background complexity, and wave white-capping interacts with sun glare to increase the proportion of saturated pixels in an image which can confuse background pixels and wildlife targets. Additional targeted imagery collection will occur in order to expand training, validation, and testing datasets to encompass the

broader range of conditions encountered during operational surveys covering large geographic areas [55]. Further, methods will be developed to assess the significance of environmental conditions on model performance.

Filtering, or sub-setting, large sets of imagery based on predicted bird occurrence is an important, practical addition to our workflow that reduces data volume and the computing resources needed for in-flight data processing. False negative filter predictions (i.e., images that contain birds but are predicted to be devoid of targets) affect the accuracy of population estimates by eliminating images containing birds from subsequent detection, classification, and counting workflow stages. Previous efforts that utilized automated methods to filter large amounts of imagery based on predicted occurrence of marine animals for the sake of processing efficiency documented data loss [56] due to false negative filter predictions. The effects of false negatives during the filtering stage could be particularly extreme for rare species. False positive filter predictions (e.g., instances when an image contains no targets but is labeled as having a target) are of lesser concern for population estimations, as these images will subsequently be passed through individual detection, classification, and counting stages where filter errors of commission may be corrected. However, false positives do increase computational demands in flight. Wildlife managers and machine learning practitioners should seek to parameterize models to obtain a satisfactory balance between processing efficiency and false negative predictions. Regardless of how this trade-off is negotiated, we recommend that all imagery be retained to estimate the effects of filter errors and to support as yet unanticipated applications. Consistent with these recommendations, Normandeau Associates, Inc. and APEM, Inc. [56] used an algorithm to filter aerial imagery based on predicted wildlife occurrence followed by manual image interpretation to identify, annotate, and count wildlife targets. They subsequently re-examined 10% of the images excluded from initial manual interpretation by the filtering algorithm and assessed bias in population estimation due to false negative filter predictions [56]. In summary, trade-offs between filter performance and processing efficiency, in particular for in-flight workflows, must consider survey and management objectives, computing resources available, and staffing resources available for post-flight bias assessment.

One significant challenge that requires additional work is the issue of double-counting bird instance predictions due to the cropping of image patches during processing and overlap in adjacent patches. This cropping is necessary for computational efficiency, however, overlap in cropped patches from a single image can result in the same bird generating a predicted instance in more than one patch. We addressed overlap through non-maximum suppression, however, additional work is needed to remove duplicate predictions from adjacent and overlapping cropped patches [57]. Operationally, addressing the issue of duplicate instances is equally important in dealing with image overlap along a flight line or in the cross-track overlap areas of images from multi-sensor arrays, if synoptic coverage of sample units is desired.

The filter and detection algorithms presented here were trained against a suite of hunted sea duck species. However, the techniques are transferable to other taxa and these models will be re-trained as additional annotated training data is generated from other imagery collections containing marine wildlife. As taxa that vary widely in body size (e.g., whales versus birds) are incorporated, additional post-processing size thresholds could be implemented to reduce over-prediction of each respective species by detection models. Implementation of the classification stage in the proposed workflow should reduce the effect of detection over-predictions on species-specific counts.

We have not yet implemented the classification (i.e., species identification) stage in our proposed workflow, but plan to pursue this in subsequent work. Our annotated training set was developed with a high degree of granularity designed to capture variation in plumage and posture within a species to maximize its utility during classification. The long-tailed distribution

apparent in our initial, limited training set, where a few dominant classes contain most instances and rare classes are represented by few instances, is common to multi-species wildlife surveys. As we implement the classification stage and incorporate additional taxa, it will be important to consider the challenge of imbalanced training data so that the classification models produce acceptable results across both abundant and rare classes [58].

Finally, we note that to obtain the high-resolution imagery used in this study the aircraft had to be operated in close proximity to the water, undermining the goal of improved safety. However, crew size was reduced from three to two, and improvements in sensor technology are rapidly eliminating this obstacle and enabling collection of imagery of similar GSD at higher, safer altitudes.

Rapid advances in remote sensing methods and artificial intelligence technologies show great promise in improving the safety and accuracy of wildlife population surveys. Automation is critical to making remote sensing wildlife surveys of broad geographic areas both time- and cost-efficient, while an acceptable level of accuracy is critical for meeting specific management objectives. With additional development and refinement, the three-stage deep learning workflow proposed here can be implemented to automate detection, classification, and enumeration to aid in wildlife distributional mapping and population estimation.

## Supporting information

**S1 Table. Subdivision of imagery dataset into training, validation, and test subdatasets.** The table details the subdivision of the imagery dataset used in this study into a training, a validation, and a test dataset. The training and validation datasets consist of images from the Nantucket Shoals study area, while the test dataset consists of images from Lake Michigan. An additional 69 images from the Lake Michigan study area, independent from the test dataset are not listed in the table, were used for further performance evaluation of the binary classifier algorithm.
(PDF)

**S2 Table. Additional test subdataset for image filtering.** An additional 69 images from the Lake Michigan study area, independent from the test dataset, were used together with 9 images from test dataset for further performance evaluation of the binary classifier algorithm.
(PDF)

**S1 File. Metadata for the shapefile used for the political boundaries in Fig 1.** The source information for the map data is covered by the Geogratis License Agreement for Unrestricted Use of Digital Data. No base map or other copyrighted material was used to create the map figure.
(PDF)

## Acknowledgments

We thank U.S. Fish and Wildlife personnel Sarah Yates for assistance in annotating imagery and Walt Rhodes III for work to standardize annotation classes. Jennifer Dieck, U.S. Geological Survey, provided administrative support while Larry Robinson assisted with image processing. We thank Colleen Anderson for editorial assistance and Steve Houdek for comments that improved an earlier draft of the manuscript. We thank Tim Ball of Fireball International Services Corporation for technical and administrative support. The findings and conclusions in this article are those of the author(s) and do not necessarily represent the views of the U.S. Fish and Wildlife Service. Findings and conclusions do represent the views of the U.S. Geological

Survey. Mention of trade names or commercial products does not constitute endorsement or recommendation for use by the U.S. Government.

## Author Contributions

**Conceptualization:** Stella X. Yu, Mark D. Koneff, David L. Fronczak, Brian R. Lubinski, Timothy P. White.

**Data curation:** Tsung-Wei Ke, David L. Fronczak, Luke J. Fara, Enrika J. Hlavacek.

**Formal analysis:** Tsung-Wei Ke, Stella X. Yu, Travis J. Harrison, Kyle L. Landolt.

**Funding acquisition:** Stella X. Yu, Mark D. Koneff, Timothy P. White.

**Investigation:** Tsung-Wei Ke, Stella X. Yu, Mark D. Koneff, David L. Fronczak, Luke J. Fara, Travis J. Harrison, Kyle L. Landolt, Enrika J. Hlavacek, Brian R. Lubinski, Timothy P. White.

**Methodology:** Tsung-Wei Ke, Stella X. Yu, Mark D. Koneff, David L. Fronczak, Luke J. Fara, Travis J. Harrison, Kyle L. Landolt, Enrika J. Hlavacek, Brian R. Lubinski, Timothy P. White.

**Project administration:** Stella X. Yu, Mark D. Koneff, Enrika J. Hlavacek, Timothy P. White.

**Resources:** Stella X. Yu, Mark D. Koneff, Kyle L. Landolt, Enrika J. Hlavacek, Timothy P. White.

**Software:** Tsung-Wei Ke, Stella X. Yu, Kyle L. Landolt.

**Supervision:** Stella X. Yu, Mark D. Koneff, Enrika J. Hlavacek, Timothy P. White.

**Validation:** Tsung-Wei Ke, Stella X. Yu, Mark D. Koneff, David L. Fronczak, Luke J. Fara, Travis J. Harrison, Kyle L. Landolt, Enrika J. Hlavacek, Brian R. Lubinski, Timothy P. White.

**Visualization:** Tsung-Wei Ke, Stella X. Yu, Luke J. Fara.

**Writing – original draft:** Tsung-Wei Ke, Stella X. Yu, Mark D. Koneff, David L. Fronczak, Luke J. Fara, Travis J. Harrison, Kyle L. Landolt, Enrika J. Hlavacek, Brian R. Lubinski, Timothy P. White.

**Writing – review & editing:** Tsung-Wei Ke, Stella X. Yu, Mark D. Koneff, David L. Fronczak, Luke J. Fara, Travis J. Harrison, Kyle L. Landolt, Enrika J. Hlavacek, Brian R. Lubinski, Timothy P. White.

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
