## [Decision Letter · Decision Letter 0]

23 May 2021

PONE-D-21-04667

Deep learning workflow to support in-flight processing of digital aerial imagery for wildlife population surveys

PLOS ONE

Dear Dr. Ke,

Thank you for submitting your manuscript to PLOS ONE. After careful consideration, we feel that it has merit but does not fully meet PLOS ONE’s publication criteria as it currently stands. Therefore, we invite you to submit a revised version of the manuscript that addresses the points raised during the review process.

According to the  PLOS ONE’s publication criteria , not on the novelty, I would like to recommend "Major Revision" by condersing the following concens,

1) some papers related to this topic would be of great interest for many of its potential readers, including the following examples: (1) Quantification of left ventricle via deep regression learning with contour guidance, IEEE Access 7: 47918-47928 (2019)   (2) Automatic Segmentation of the Femur and Tibia Bones from X-ray Images Based on Pure Dilated Residual U-Net, Inverse Problems and Imaging, 10.3934/ipi.2020057 (3)GVFOM: a novel external force for active contour based image segmentation, Information Sciences 506 (2020) 1–18 (4) Automatic segmentation of the cardiac MR images based on nested fully convolutional dense network with dilated convolution, Biomedical Signal Processing and Control, Volume 68, July 2021, https://doi.org/10.1016/j.bspc.2021.102684

2)  Annotation for the training dataset should be paraphrased more since it will affect the model and the results.

3) When it comes to the experiments, *the cross-validation experiments for the different areas should be employed for validation and the MASK RCNN with and without classification should be taken into account.*

*4) pls refer to the reviewers' other messages for revision.*

We look forward to receiving your revised manuscript.

Kind regards,

Yuanquan Wang

Academic Editor

PLOS ONE

Journal Requirements:

I n your Methods section, please provide additional information regarding the permits you obtained for the work. Please ensure you have included the full name of the authority that approved the field site access and, if no permits were required, a brief statement explaining why.

We note that you have stated that you will provide repository information for your data at acceptance. Should your manuscript be accepted for publication, we will hold it until you provide the relevant accession numbers or DOIs necessary to access your data. If you wish to make changes to your Data Availability statement, please describe these changes in your cover letter and we will update your Data Availability statement to reflect the information you provide.

4. We note that Figure 1 in your submission contain map images which may be copyrighted. All PLOS content is published under the Creative Commons Attribution License (CC BY 4.0), which means that the manuscript, images, and Supporting Information files will be freely available online, and any third party is permitted to access, download, copy, distribute, and use these materials in any way, even commercially, with proper attribution. For these reasons, we cannot publish previously copyrighted maps or satellite images created using proprietary data, such as Google software (Google Maps, Street View, and Earth). For more information, see our copyright guidelines: http://journals.plos.org/plosone/s/licenses-and-copyright.

Reviewers' comments:

Reviewer's Responses to Questions

**Comments to the Author**

1. Is the manuscript technically sound, and do the data support the conclusions?

Reviewer #1: Partly

Reviewer #2: Partly

2. Has the statistical analysis been performed appropriately and rigorously? 

Reviewer #1: No

Reviewer #2: Yes

3. Have the authors made all data underlying the findings in their manuscript fully available?

Reviewer #1: No

Reviewer #2: No

4. Is the manuscript presented in an intelligible fashion and written in standard English?

Reviewer #1: Yes

Reviewer #2: Yes

5. Review Comments to the Author

Reviewer #1: The major theme of this paper is to evaluate the performance of deep learning models for monitoring of wildlife populations. Two core techniques are involved, including 1) a binary classifier algorithm as a means to filter images based on predicted bird occurrence and 2) a Mask R-CNN algorithm for detecting individual marine bird instances.

On the whole, the application project of the paper is interesting, but the novelty in the methodology is rather limited. In a great sense, the paper merely applies existing methods/algorithms to the specific application field. On my side, it should not be submitted as a research paper, and hence I cannot give positive assessment on it.

Reviewer #2: The authors proposed a workflow for wildlife population surveys. In this work, they designed algorithms for imagery filtering and bird detection. It is an effective way to apply the classification model before detection. However, there are also some points that should be addressed:

1. Did authors apply multiple annotation for the training dataset? The accurateness of the annotation provides not only the guarantee of the training and testing authenticity, but also the upper boundary of the trained model.

2. It is great to use different survey areas to evaluate the robustness of the model. For better validation, it is encouraged to apply the cross-validation experiments for the different areas, such as training on images from the Lake Michigan and testing on images from the Nantucket Shoals, training on images from both two areas and testing from both two areas, which can fully report the superiority of the proposed method.

3. How did authors choose the probability threshold in the first stage?

4. The authors compared the MASK R-CNN with classification model in the first stage. However, I think it is better to apply the comparisons between MASK R-CNN without classification model and MASK R-CNN with classification model.

5. For the results of detection, did author consider the deviation/error of the first stage, e.g., classification model?

6. PLOS authors have the option to publish the peer review history of their article (what does this mean?). If published, this will include your full peer review and any attached files.

Reviewer #1: No

Reviewer #2: No

---

## [Author Response · Author response to Decision Letter 0]

4 Aug 2022

[R1. Lack of novelty]

We clarify that this paper does not aim at proposing new classification / detection algorithms, but an automated workflow to support large-scale wildlife surveys. Endowed by the great success of deep learning algorithms, we alleviate the painful need of storing and manually labeling large amounts of in-flight imagery. We combine the state-of-the-art classification and detection methods for filtering images with any bird occurrence and detecting the bird instances correspondingly. In this paper, our main focus is to provide some insights of deep learning algorithms to address issues of data processing in wildlife surveys.

[R2. Multiple annotation for the imagery]

In this paper, we explore such prototype models for automating data processing and do not include multiple annotations for the imagery. But this is a good suggestion and we are in agreement regarding the utility of this approach. We are developing an operational annotation and training data development workflow in which at least 2 annotators label the same set of imagery, with a process for arbitrating annotation discrepancies through a third species expert. This will allow us to better assess and control annotation data quality.

[R2. Cross-validating Mask R-CNN]

We agree that this is a good suggestion. We updated the cross-validation experiments in the manuscript.

[R2. Probability threshold selection for filtering]

In this paper, we simply select the probability threshold based on the performance over 78 additional images from Lake Michigan. We have not yet determined the specific threshold that we will use operationally and realize that the threshold may vary among surveys given specific survey objectives. Operationally, our workflow will incorporate some sampling of the images which do not pass the filter to manually review them and determine evaluate errors of omission. In a related effort we are working to develop statistical population estimation frameworks that incorporate machine-learning derived outputs in a rigorous manner. Errors of commission, passing imagery with no targets to subsequent workflow stages, primarily affect the in-flight performance and computational burden, and are less of our concern.

[R2. Compare Mask R-CNN with/without a classifier for filtering]

In the first stage of image filtering, we do not consider the fine-grained categories of bird instances, but only focus on detecting any bird occurrence. We, therefore, disagree with the recommendation to compare Mask R-CNN with and without a classifier for image filtering. We do agree that it would be a good idea to compare Mask R-CNN with / without a classifier for the third stage. In our future work, we could conduct comparison between two different settings: 1) detection using the Mask R-CNN without a classifier and categorization using additional classification model, 2) detection and categorization using the Mask R-CNN with a classifier.

[R2. Compute the deviation of the first stage]

If we are understanding the comment correctly, no we did not evaluate the effects of error in the first stage classifier(filter) downstream in individual detection or classification stages. Our primary concern is how machine-learning outputs and errors at each stage in our workflow propagate as ecologically meaningful uncertainty in wildlife population estimates. While outside the scope of this paper, we are, therefore, focused questions about how machine-learning processes and outputs integrate within standard statistical frameworks for wildlife population size estimation. We envision, and are investigating, series of linked submodels that propagate error at all stages of the workflow (annotation, filter, detection, classification) into ecologically interpretable estimates of uncertainty in population size.

---

## [Decision Letter · Decision Letter 1]

20 Jun 2023

Deep learning workflow to support in-flight processing of digital aerial imagery for wildlife population surveys

PONE-D-21-04667R1

Dear Dr. Ke,

We’re pleased to inform you that your manuscript has been judged scientifically suitable for publication and will be formally accepted for publication once it meets all outstanding technical requirements.

Kind regards,

Judi Hewitt

Academic Editor

PLOS ONE

Additional Editor Comments (optional):

Reviewers' comments:

Reviewer's Responses to Questions

**Comments to the Author**

1. If the authors have adequately addressed your comments raised in a previous round of review and you feel that this manuscript is now acceptable for publication, you may indicate that here to bypass the “Comments to the Author” section, enter your conflict of interest statement in the “Confidential to Editor” section, and submit your "Accept" recommendation.

Reviewer #3: All comments have been addressed

2. Is the manuscript technically sound, and do the data support the conclusions?

Reviewer #3: Yes

3. Has the statistical analysis been performed appropriately and rigorously? 

Reviewer #3: Yes

4. Have the authors made all data underlying the findings in their manuscript fully available?

Reviewer #3: Yes

5. Is the manuscript presented in an intelligible fashion and written in standard English?

Reviewer #3: Yes

6. Review Comments to the Author

Reviewer #3: To support in-flight orthorectification and machine learning processing to detect and classify wildlife from imagery in near real-time, the authors evaluated deep learning methods that address hardware limitations and the need for processing efficiencies to support the envisioned in-flight workflow.

This paper has high application value.

But the novelty is limited.

7. PLOS authors have the option to publish the peer review history of their article (what does this mean?). If published, this will include your full peer review and any attached files.

Reviewer #3: No

---

## [Editor Report · Acceptance letter]

11 Mar 2024

PONE-D-21-04667R1 

PLOS ONE

Dear Dr. Ke, 

I'm pleased to inform you that your manuscript has been deemed suitable for publication in PLOS ONE. Congratulations! Your manuscript is now being handed over to our production team.

Kind regards, 

on behalf of

Dr. Judi Hewitt 

Academic Editor

PLOS ONE